# Immiscible hydrous Fe–Ca–P melt and the origin of iron oxide-apatite ore deposits

Tong Hou [1,2], Bernard Charlier [3], François Holtz[1], Ilya Veksler[4,5], Zhaochong Zhang[2], Rainer Thomas[4] & Olivier Namur[1,6]

The origin of iron oxide-apatite deposits is controversial. Silicate liquid immiscibility and separation of an iron-rich melt has been invoked, but Fe–Ca–P-rich and Si-poor melts similar in composition to the ore have never been observed in natural or synthetic magmatic systems. Here we report experiments on intermediate magmas that develop liquid immiscibility at 100 MPa, 1000–1040 °C, and oxygen fugacity conditions ($fO_2$) of $\Delta$FMQ = 0.5–3.3 (FMQ = fayalite-magnetite-quartz equilibrium). Some of the immiscible melts are highly enriched in iron and phosphorous ± calcium, and strongly depleted in silicon (<5 wt.% $SiO_2$). These Si-poor melts are in equilibrium with a rhyolitic conjugate and are produced under oxidized conditions (~FMQ + 3.3), high water activity ($aH_2O \geq 0.7$), and in fluorine-bearing systems (1 wt.%). Our results show that increasing $aH_2O$ and $fO_2$ enlarges the two-liquid field thus allowing the Fe–Ca–P melt to separate easily from host silicic magma and produce iron oxide-apatite ores.

[1] Institute of Mineralogy, Leibniz Universtät Hannover, 30167 Hannover, Germany. [2] State Key Laboratory of Geological Process and Mineral Resources, China University of Geosciences, 100083 Beijing, China. [3] Department of Geology, University of Liege, 4000 Sart Tilman, Belgium. [4] GFZ German Research Center for Geosciences, Telegrafenberg, 14473 Potsdam, Germany. [5] Geological Department, Perm State University, Bukireva 15, Perm, Russia 614990. [6] Department of Earth and Environmental Sciences, KU Leuven, 3001 Leuven, Belgium. Correspondence and requests for materials should be addressed to T.H. (email: thou@cugb.edu.cn)

The origin of orebodies composed of low-Ti iron oxide minerals (magnetite and/or hematite) and apatite in (sub) volcanic rocks is controversial[1–12]. These rocks, essentially free of silicates and sufficiently enriched in Fe to be recoverable, have been classified as Kiruna-type or iron oxide-apatite (IOA) deposits[1–3]. Their enrichment in Fe and P has been variously attributed to magmatic and hydrothermal ore-forming processes. Metasomatic replacement of the host igneous rocks by convecting fluids[4,6], proposed as a likely mechanism due to the pervasive hydrothermal alteration of the ore, is supported by the low Ti content and trace element characteristics of magnetite crystals[4,9,10]. Alternatively, IOA deposits may represent volcanic flows or shallow magma intrusions as suggested by several field relationships including discordant veins and dykes of magnetite-apatite ores intruding their host rocks, magma flow structures, vesicular textures, and volcanic bombs[5,7]. In this case, the formation of Fe-rich and P-rich rocks might be explained by liquid immiscibility and segregation of a Fe–P-rich immiscible magma from its rhyolitic counterpart[7]. The development of immiscibility is supported by the coexistence of two types of melts in glassy matrices and inclusions hosted by phenocrysts in the ore and in andesitic wall rocks[7,11,13,14]. However, none of these immiscible melts have compositions representative of IOA ores. Experimental evidence for the formation of such silica-poor iron oxide melts at magmatic conditions is also lacking[15,16].

Evolved basaltic magmas can split into immiscible rhyolitic (dacitic) and ferrobasaltic melts along their crystallization path at temperatures below 1040–1020 °C[17–19]. $P_2O_5$ in the bulk composition promotes the development of silicate liquid immiscibility and this oxide strongly concentrates in the Fe-rich melt[16,20,21]. Experimental and natural Fe-rich immiscible melts generally contain 35–45 wt.% $SiO_2$ and only a few wt.% $P_2O_5$[18,19,21–28].

Silicate phases predominantly crystallize from such melts[18,19], producing oxide-apatite gabbros of moderate economic interest[29,30]. Extreme enrichment of apatite and iron oxide over silicate minerals, as observed in IOA deposits, cannot simply result from differential crystal settling in an iron-rich silicate melt. This is because, with the exception of plagioclase, common silicate minerals (actinolite and diopside) are denser than the melt and would sink along with the oxides. A more efficient mechanism for the production of IOA deposits would be direct crystallization of a Fe–P-rich and Si-depleted magma.

Here, we provide an original solution to this challenging issue based on results obtained from experiments performed in realistic conditions of pressure and temperature in an internally heated pressure vessel (IHPV). We used experimental starting material which was prepared from a series of mixtures between two mafic end-members and a rhyolitic composition (Supplementary Fig. 1 and Supplementary Tables 1, 2). We show that liquid immiscibility develops in the intermediate magmas at conditions relevant to the magmatic reservoirs of most subvolcanic IOA deposits ($P = 100$ MPa, $T = 1000$–1040 °C). With elevation of oxygen fugacity and water activity, nearly pure Fe–Ca–P melts that are compositionally identical to typical IOA ores are produced by liquid immiscibility. This finding allows us to conclude that liquid immiscibility is the key process in the formation of IOA deposits. This is extremely important for the establishment and refinement of a petrogenetic model for IOA ores.

## Results

**Phase equilibria and immiscibility textures.** Experimental conditions and phase assemblages are summarized in Supplementary Table 3. All run products contain crystal phases and either a

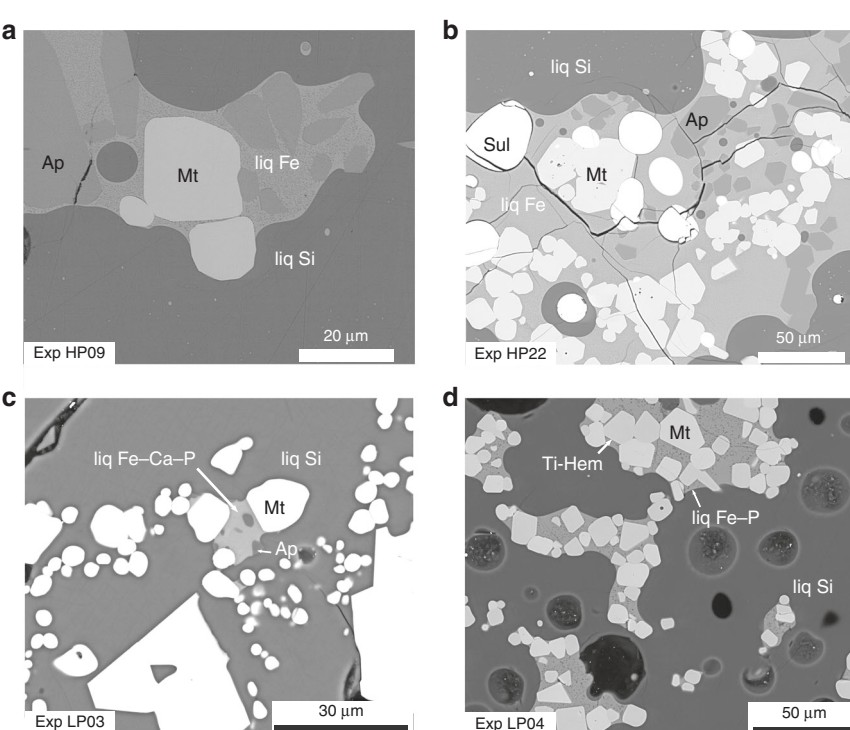

**Fig. 1** Back-scattered electron images of selected experiments showing liquid immiscibility between Fe-rich and Si-rich glass. **a**, **b** Typical irregularly shaped (coalesced) patches of Fe-rich silicate glass (liq Fe) within Si-rich glass (liq Si). Magnetite and/or apatite are preferentially enclosed in the immiscible Fe-rich silicate glasses. **c** Fe-Ca-P glass (liq Fe–Ca–P) separated from the Si-rich glass (liq Si). Magnetite and apatite are crystalline phases in both liquids. **d** Irregularly shaped (coalesced) patches of Fe-P glass (liq Fe–P) within Si-rich glass. Oxide minerals (Ti-rich hematite and magnetite) are predominantly hosted by the Fe–P glass. Abbreviations: Mt, magnetite; Ti-Hem, solid solution of ilmenite and hematite; Ap, apatite; Sul, sulfide; liq Fe, Fe-rich silicate glass; liq Fe–Ca–P, Fe-Ca-P glass; liq Fe–P, Fe-P glass; liq Si, Si-rich glass

single homogenous melt or two distinct immiscible melts quenched to glass. Solid phases are magnetite, apatite, fayalite (or fayalitic olivine), a silica phase (tridymite), and occasionally titano-hematite and clinopyroxene. A single homogeneous melt is found in some experiments with high bulk $P_2O_5$ contents (1.1–2.3 wt.% $P_2O_5$; Supplementary Table 2), indicating that, despite the critical role of phosphorus on the development of liquid immiscibility[31], other compositional parameters must contribute significantly to the onset of unmixing. We note that a single melt is also observed in experiments performed at the highest temperature (i.e., 1040 °C) suggesting that in our multi-component system the apex of the binodal lies beneath 1040 °C, as already identified in dry ferrobasalts[19]. We also note that all experiments performed below 1040 °C under oxidizing conditions (fayalite-magnetite-quartz equilibrium) (~FMQ + 3) developed immiscibility while some experiments performed at identical temperature under more reduced conditions do not show immiscibility.

Experimental products with distinct immiscibility typically show sharp two-liquid interfaces (Fig. 1). Immiscible melts form globules or domains of various sizes (including nano-scale droplets). We observe no compositional difference between small and large melt pools in individual experiments, suggesting complete equilibration of the two melts. In runs with sufficiently large globules, the Fe-rich melt droplets display very small wetting angles with magnetite, apatite, and fayalite, and these phases form euhedral crystals preferentially concentrated in the Fe-rich melt (Fig. 1). Experiments in which we added an FeS component (HP22–27) also contain large spherical or ovoid droplets of sulfide melt dispersed in the silicate glasses: our experimental products therefore contain three immiscible liquids (Fig. 1b).

**Olivine and oxide mineral compositions**. Electron microprobe analyses of the crystal phases are presented in Supplementary Data 1. Olivine compositions vary from $Fo_{24}$ to $Fo_2$ (Fo = 100 [Mg/(Mg + $Fe^{2+}$)]) with decreasing $fO_2$ and temperature. Under oxidizing conditions (FMQ + 3.1 to FMQ + 3.3), experiments contain two oxide minerals (Fig. 1d), a rhombohedral oxide of the hematite-ilmenite solid solution (14.84–24.64 wt.% $TiO_2$) and magnetite (0.38–1.53 wt.% $TiO_2$). Under more reducing

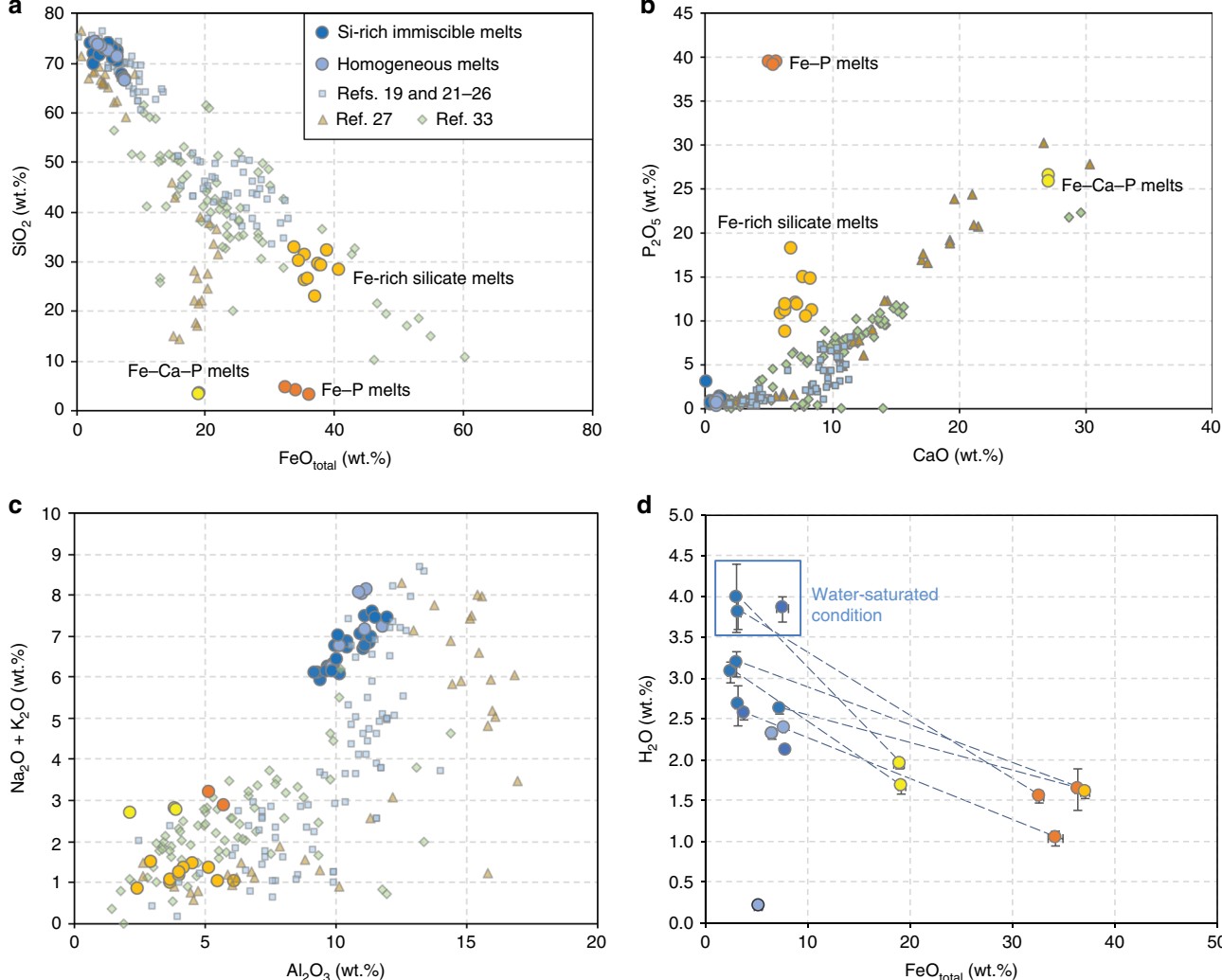

**Fig. 2** Compositional variations of the experimental melts: **a** $FeO_{tot}$ vs. $SiO_2$; **b** CaO vs. $P_2O_5$; **c** $Al_2O_3$ vs. $Na_2O + K_2O$; and **d** $H_2O$ vs. $FeO_{tot}$. Three types of Fe-rich immiscible melts were produced: Fe-rich silicate melts and Fe-P and Fe-Ca-P melts. Literature data are plotted for comparison: Fe-rich melts in the mesostasis of natural samples[33], mineral-hosted melt inclusions[27], and experimental immiscible melts in tholeiitic systems[19, 21-26]. Error bars represent one standard deviation of replicate analyses

**Table 1 Compositional ranges of melts in this study**

| Liquid pairs | SiO$_2$ | TiO$_2$ | Al$_2$O$_3$ | FeO$_{tot}$ | MnO | MgO |
|---|---|---|---|---|---|---|
| Fe-rich silicate melts | 22.69–32.66 | 0.81–2.82 | 2.44–6.14 | 33.85–40.84 | 0.99–2.73 | 1.48–4.21 |
| Si-rich conjugates | 67.63–72.94 | 0.20–0.56 | 9.65–11.36 | 4.53–7.21 | 0.13–0.27 | 0.20–0.55 |
| Fe-P melts | 3.09–4.55 | 0.99–1.53 | 2.18–3.90 | 32.63–36.36 | 2.99–3.28 | 4.87–5.22 |
| Si-rich conjugates | 70.97–73.71 | 0.54–0.60 | 9.22–9.69 | 2.48–3.20 | 0.06–0.12 | 0.12–0.16 |
| Fe-Ca-P melts | 3.33–3.40 | 0.98 | 5.19–5.72 | 18.88–19.15 | 3.53–5.66 | 7.23–7.48 |
| Si-rich conjugates | 73.23–74.13 | 0.68–0.73 | 9.42–9.98 | 2.99–3.05 | 0.15–0.21 | 0.25–0.33 |
| Si-rich melts with nano-scale Fe-rich globules | 66.55–73.90 | 0.26–0.52 | 9.85–11.98 | 3.73–7.74 | 0.05–0.27 | 0.05–0.62 |
| Homogeneous Si-melts (no immiscibility) | 66.43–74.16 | 0.24–0.35 | 9.90–11.80 | 3.13–7.70 | 0.14–0.35 | 0.27–0.62 |
| **Liquid pairs** | **CaO** | **Na$_2$O** | **K$_2$O** | **P$_2$O$_5$** | **F** | |
| Fe-rich silicate melts | 6.02–8.49 | 0.57–0.83 | 0.25–0.69 | 8.55–18.07 | 0.23–0.41 | |
| Si-rich conjugates | 1.00–1.40 | 1.73–2.48 | 4.17–4.99 | 0.44–1.09 | 0.10–0.17 | |
| Fe-P melts | 5.13–5.74 | 1.74–1.90 | 0.77–1.00 | 39.02–39.40 | 0.48–0.60 | |
| Si-rich conjugates | 0.15–0.21 | 1.47–1.86 | 3.95–4.23 | 2.84–2.96 | 0.31–0.69 | |
| Fe-Ca-P melts | 27.16–27.20 | 2.19–2.50 | 0.65–0.69 | 25.71–26.43 | 2.51–2.71 | |
| Si-rich conjugates | 0.52–0.74 | 1.69–2.20 | 4.21–4.54 | 0.37–0.62 | 0.15–0.17 | |
| Si-rich melts with nano-scale Fe-rich globules | 0.51–1.37 | 1.92–2.43 | 4.20–5.16 | 0.30–1.00 | 0.10–0.19 | |
| Homogeneous Si-melts (no immiscibility) | 0.95–1.19 | 2.08–3.29 | 4.15–4.89 | 0.20–0.89 | 0.09–0.17 | |

conditions (FMQ + 0.5), the oxide phase is magnetite (Supplementary Data 1).

**Melt compositions.** Experimental melt compositions are reported in Supplementary Data 1 and illustrated in Harker diagrams (Fig. 2). Depending on bulk compositions and experimental conditions, we observe three types of sulfur-free Fe-rich immiscible melts defining a broad compositional range: a Fe-rich silicate melt similar to a ferrobasalt, a Fe–P melt, and a Fe–Ca–P melt (Fig. 2; Table 1). Fe-rich silicate melts contain 22.7–32.7 wt.% SiO$_2$, 33.9–40.8 wt.% FeO$_{tot}$, 8.6–18.1 wt.% P$_2$O$_5$, and are enriched in MgO, CaO, and TiO$_2$. Most Fe-rich silicate melts were produced under nominally dry conditions (Supplementary Table 3). Fe–P melts were produced under relatively oxidizing (FMQ + 3.1 to FMQ + 3.3) and hydrous conditions ($a$H$_2$O = 0.7–1.0). They are homogenous and contain 32.6–33.4 wt.% FeO$_{tot}$, 39.0–39.4 wt.% P$_2$O$_5$, with minor SiO$_2$ (3.1–4.6 wt.%), TiO$_2$ (1.0–1.5 wt.%), Al$_2$O$_3$ (2.2–3.9 wt.%), MgO (4.9–5.2 wt.%), and CaO (5.1–5.7 wt.%). Fe–Ca–P melts, produced under water-saturated and oxidizing conditions (FMQ + 3.2 to FMQ + 3.3), contain 18.4–19.2 wt.% FeO$_{tot}$, 27.2–33.4 wt.% CaO, 23.8–26.4 wt.% P$_2$O$_5$, 7.2–7.5 wt.% MgO, minor SiO$_2$ (3.3–5.1 wt.%), TiO$_2$ (0.4–1.0 wt.%), and Al$_2$O$_3$ (5.2–5.7 wt.%), and have relatively high F contents (2.5–2.7 wt.%).

Conjugate Si-rich immiscible melt compositions vary from dacite to rhyolite, and are compositionally similar to felsic rocks hosting some IOA deposits (Fig. 3)[32–37]. They contain 67.6–74.1 wt.% SiO$_2$, 9.2–11.4 wt.% Al$_2$O$_3$, 2.5–7.7 wt.% FeO$_{tot}$, 4.0–5.0 wt.% K$_2$O, and 1.5–2.5 wt.% Na$_2$O. The Si-rich immiscible melts equilibrated with Fe–P and Fe–Ca–P melts contain relatively little CaO (0.1–0.2 and 0.5–0.7 wt.%, respectively), whereas those equilibrated with Fe-rich silicate melts contain 1.0–1.4 wt.% CaO (Table 1). Silicate melts in experiments without liquid immiscibility are rhyolitic, and do not differ significantly from those coexisting with a Fe-rich liquid, suggesting that they are close to the binodal surface (Supplementary Data 1). Sulfide droplets have compositions close to stoichiometric FeS.

In water-saturated experiments, Si-rich melts contain more than 3.5 wt.% H$_2$O, whereas the Fe-rich conjugates usually contain less than 2 wt.% H$_2$O (Fig. 2d). This indicates that water preferentially partitions into the Si-rich liquids as also observed in water-unsaturated experiments. Volatile element distributions between the conjugate melts are described using partition

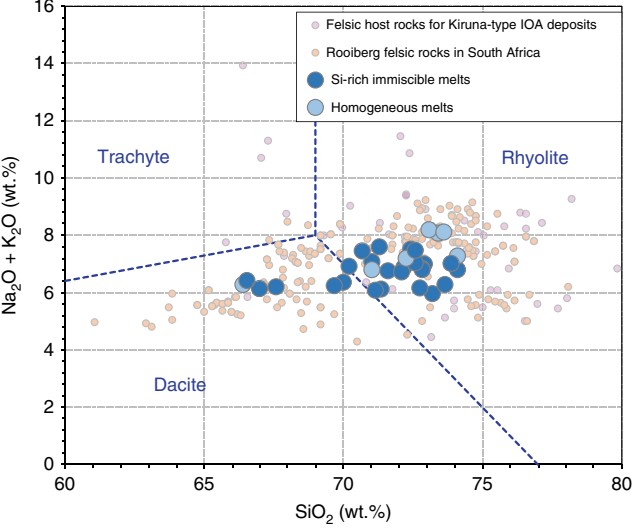

**Fig. 3** TAS diagram showing the experimental dacitic and rhyolitic Si-rich melts. These immiscible Si-rich melts are similar to natural felsic rocks hosting Kiruna-type IOA deposits[34–37] and those of the Rooiberg Group in South Africa[38]

coefficients defined as $D_i = C_i^{LFe}/C_i^{LSi}$, where $C$ represents the concentration (wt.%) of component $i$ in the Fe-rich (LFe) and Si-rich (LSi) conjugate liquids. Liquid–liquid $D_{H_2O}$ values range from 0.39 to 0.69. F and SO$_3$ are enriched in the Fe-rich melts. $D_F$ is about 2. This is consistent with our previous study in the F-rich multicomponent system[28], but contrasts with experimental results in simplified systems[16] in which F was reported to partition nearly equally between the mafic and silicate liquids ($D_F$ = 1 ± 0.6). Due to the differences of the degree of melt polymerization in the immiscible conjugates[38], SO$_3$ partitions preferentially into the Fe-rich melts (Fe-rich melt: 1.07–1.53 wt.% SO$_3$; $D_{SO_3}$: 21.5–38.3) and sulfur concentrations in the Si-rich immiscible melts are therefore very low (0.03–0.06 wt.%).

**Discussion**
Several experimental studies in multi-component systems have reported immiscible melts. In most cases, the Fe-rich melts have

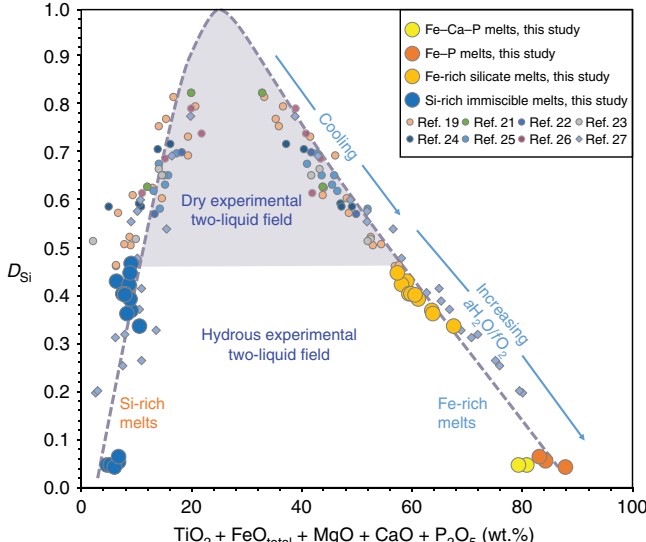

**Fig. 4** Compositions of conjugate immiscible melts in this study compared to previous experimental melt pairs in tholeiitic systems[19, 21–26] and native iron-hosted melt inclusions in lavas of the Siberian Traps[27]. $D_{Si}$, the partitioning of $SiO_2$ between the Fe-rich and Si-rich melts, expresses the compositional gap between the two melts, plotted as a function of elements partitioned into the Fe-rich melts

at least 30 wt.% $SiO_2$[19,21–26] (Fig. 4). With cooling, immiscible melt pairs become increasingly contrasted in composition, but dry Fe-rich melts reported so far have never approached the extreme, Si-free, composition of IOA ores (predominantly iron, calcium, phosphorous, with minor magnesium and titanium). The most extreme Fe–Ca–P-rich and Si-poor compositions reported to date (15 wt.% $SiO_2$; 22 wt.% $FeO_{tot}$; 27 wt.% CaO; 5–30 wt.% $P_2O_5$) have been observed in native iron-hosted immiscible melt pools from the Siberian Traps for which the conditions of formation are uncertain[27]. They are, however, known to have equilibrated at extremely reduced redox conditions (close to IW) which are unrealistic for the formation of IOA deposits. These inclusions nevertheless indicate that Fe-rich immiscible melts in natural systems may become extremely enriched in Fe, Ca, and P, and depleted in Si (Fig. 4), although the compositions observed in Siberia still do not match the compositions of IOA ore deposits.

At the most reducing conditions investigated in our study, FMQ + 0.5, the dry Fe-rich immiscible melts produced in our experiments contain 23–33 wt.% $SiO_2$ (Table 1). These compositions are comparable to native iron-hosted melt inclusions but were produced at redox conditions more relevant to the formation of IOA ore deposits. With increasing $aH_2O$, immiscible pair compositions become more contrasted (Fig. 4). In water-saturated experiments, we also observe high $P_2O_5$ contents in Fe-rich silicate immiscible melts (up to 18.07 wt.%), which indicates that addition of $H_2O$ into magmatic systems of intermediate composition enhances the development of immiscibility and expands the width of the binodal surface compared to that in dry systems (Fig. 4). We believe that even more contrasted compositions than those reported in this study could be produced at temperatures <1000 °C.

At more oxidizing (FMQ + 3.1 to FMQ + 3.3) and hydrous conditions, immiscible pair compositions are extremely contrasted compared to the liquids produced at FMQ + 0.5 and the Fe-rich melt has very little silica (Fig. 4). The major element partition coefficients between the two liquids are highly correlated with the degree of polymerization of the Si-rich melt[20]. With increasing $fO_2$, $Fe^{3+}/Fe^{2+}$ increases in the Si-rich melt, which

becomes more polymerized because $Fe^{3+}$ is a network former whereas $Fe^{2+}$ is a network modifier[39]. Consequently, immiscible pair compositions become more contrasted. In addition, the conjugate Fe-rich melt contains more $Fe^{3+}$, which substitutes for $Si^{4+}$ via the reaction $P^{5+} + Fe^{3+} \leftrightarrow 2Si^{4+}$. Thus, increasingly hydrous and oxidizing conditions can explain the formation of Fe–P-enriched silicate liquids (Fig. 4), which is consistent with the recent studies on El Laco deposit[11].

Fluorine also plays an important role on the development of liquid–liquid immiscibility[28]. Fluorine complexing with Mg in the melt[40] decreases the activity of MgO, therefore increasing the activity of FeO and favouring the development of liquid immiscibility[28]. Fe–Ca–P melts contain more F (avg. 2.6 wt.%) compared to Fe–P melts (0.6 wt.%; Supplementary Data 1). We therefore suggest that fluorine also complexes with Ca[41], such that addition of F produces Ca-enriched Fe–P melts. This implies that changing F in the parental magmas prior to immiscibility may lead to a range of Fe–P-dominated melts with variable amounts of calcium when immiscibility develops. Such a mechanism could account for contrasted amounts of apatite in different IOA deposits, i.e., from almost fluorapatite-free to the fluorapatite-rich deposits that are currently mined.

Our experiments produced several types of Fe-enriched and P-enriched immiscible melts. In particular, the composition of the Si-depleted and Fe–Ca–P-enriched melts is relevant to IOA ore production. These liquids are in equilibrium with typical dacites and rhyolites commonly observed to host IOA ores[32–37]. However, the formation of such contrasted immiscible melt pairs, that could potentially form ore deposits, requires hydrous and oxidizing environments. In contrast to tholeiitic magmatism, these conditions are to be expected for IOA ores for two main reasons. First, IOA deposits are commonly located in convergent settings where slab dehydration leads to the formation of water-bearing magmas[42]. There are also commonly observed in extensional intraplate setting where crustal rocks melting[43,44] may also produce hydrous magmas[10]. Second, the redox state of arc magmas is usually considered to be close to FMQ + 1, which is more oxidized than tholeiitic magmas[45]. Further oxidation of the magma is likely to occur during crustal assimilation and magma degassing. Indeed, systematic O-Sr-Nd isotopic studies of IOA ores suggest a significant crustal component[46]. Interactions with sediments such as carbonates or evaporites significantly oxidize magmas by decarbonatization and $CO_2$ fluxing or addition of $S^{6+}$ from assimilated gypsum[11,46]. As vapor-saturated magmas degas during ascent, the release of $H_2O$ and $CO_2$ from the silicate melt further oxidizes the magma[47,48]. Degassing of sulfur species can either oxidize or reduce the magma depending on the valence state of sulfur in the melt ($S^{2-}$ or $S^{6+}$) and the fluid phase ($H_2S$ or $SO_2$). In IOA deposits, sulfur dissolves in the melt predominantly as $S^{6+}$ as supported by the common presence of anhydrite[46]. Thus, sulfur degassing can lead to significant oxidation of the melt following the reaction $SO_4$ (melt) + 2FeO (melt) = $SO_2$ (gas) + $Fe_2O_3$ (melt) + $1/2O_2$ (melt). In addition, $SO_2$ is increasingly favored over $H_2S$ since the following equilibrium shifts to the left with decreasing pressure[49]: $SO_2 + 3H_2 = 2H_2O + H_2S$. Therefore, sulfur degassing leads to a continuous increase of $Fe^{3+}/Fe^{2+}$ in magmatic melts during decompression. The formation of IOA ore is thus intimately related to the emplacement dynamics of the host magmatic system: the parental intermediate magma is emplaced at shallow pressure, interacts with host sediments, and degasses. These processes produce oxidation of the magma that triggers immiscibility and the formation of Fe–Ca–P melt which further crystallizes to form IOA ores at the level of emplacement. In addition, crystallizing magnetite and apatite grains are preferentially wetted by the immiscible Fe-rich melt (Fig. 1). This may further enrich the crystal +

Fe-rich melt mush in elements of economic interest. Complete crystallization of the mush products leads to the formation of IOA ore.

## Methods

**Starting material synthesis**. To prepare the synthetic mafic end-members, high-purity, commercially available oxide powders ($SiO_2$, $TiO_2$, $Al_2O_3$, $Fe_2O_3$, MnO, and MgO) were mixed with ethanol in an automatic agate mortar and pestle for more than 5 h. The rhyolitic composition was prepared from high-purity oxides and carbonates, and homogenized in an agate planetary ball mill. The powder mixture was then melted in a Pt crucible at 1600 °C (atmospheric oxygen fugacity) for 3 h. The glass was then ground in a steel mortar and re-melted in the furnace (1600 °C, 3.5 h) to homogenize the material and promote complete $CO_2$ degassing. After quenching, pieces of the glass were separated, mounted in epoxy, polished, and analyzed by electron microprobe (Supplementary Table 1).

Starting compositions were prepared from a series of mixtures between two mafic end-members and a rhyolitic composition (Supplementary Fig. 1 and Table 1-2). The two mafic end-members are mixtures of fayalite and magnetite in the proportions 30:70 (M1) and 60:40 (M2). Phosphorous was added as $Ca_3(PO_4)_2$ to them in various concentrations. To simulate the volatile-rich nature of IOA deposits, 2–6 wt.% $H_2O$ was added to the starting compositions for some experiments. Because fluorapatite and sulfur-rich minerals are common in IOA deposits, all experiments contained 0.4–0.6 wt.% F, added as $CaF_2$, and three experiments contained 3 wt.% S, added as FeS (Supplementary Table 2).

**Phase equilibria experiments**. All experiments were performed at 100 MPa in large volume IHPV[50] at the Leibniz University of Hannover (Germany). Starting powders were weighed and placed in Au capsules (20 mm length and 2.8 mm internal diameter, with a 0.2 mm wall thickness). One end of each capsule was welded shut before the starting material was inserted. The other capsule end was immediately welded shut for dry samples, whereas water-bearing samples were frozen in liquid nitrogen before welding. This method minimizes water vaporization during welding. Capsules were weighed after welding and placed in a dry furnace at 150 °C for 1–2 h before being re-weighed to check if any material loss occurred. During the experiments, the capsules were fixed to a Pt-wire in the hot spot of a double-wire element furnace. Temperature was controlled using two S-type thermocouples while two additional S-type thermocouples were used to monitor the sample temperature. The temperature gradient across the sample was less than 5 °C. Samples were pressurized cold to ~80% of final pressure, then heated to the final temperature while pressure was slowly increased. Temperature was increased with a ramp of 0.8 °C/s to 30 °C below the final temperature, and then 0.3 °C/s to the final temperature (1000–1040 °C). Experiments were run for 48–168 h, then quenched by fusing the Pt-wire and dropping the capsules onto a cold (~25 °C) copper block at the bottom of the sample holder. The quench rate was ~150 °C/s. Re-weighing of the capsules after the experimental runs showed identical weights for most capsules, indicating that no volatiles were lost during the experiments. Several chips of each experimental product (about 2 mm in diameter) were prepared as polished thin sections or mounted in epoxy for observation and electron microprobe analyses.

**Oxygen fugacity calculations**. Experiments were performed at $fO_2$ ranging from FMQ + 0.5 to FMQ + 3.3, in order to investigate the effect of changing redox conditions on resulting phase equilibria.

For experiments at intrinsic $fO_2$ conditions, we used an IHPV with Ar as the sole pressure medium. The intrinsic oxygen fugacity in capsules with pure $H_2O$ fluid (mole fraction of water in the fluid $X^f_{H_2O} = 1$) in the IHPV used in this study was determined by NiPd-solid sensors[51] at 1200 °C and 200 MPa. The obtained $fO_2$ corresponds to NNO + 2.6 (±0.5; 1σ from microprobe analyses of the NiPd alloy) where NNO refers to the Ni–NiO buffer[52]. This corresponds to 3.3 log units above the FMQ solid oxygen buffer (hereafter labeled FMQ + 3.3). The $fO_2$ at $H_2O$ undersaturated conditions can be estimated using the relation $\log fO_{2capsule} = \log fO_2$ (at $aH_2O = 1$) $+ 2 \log aH_2O$[53,54], where $aH_2O$ is determined from the water concentration in the melt following the model of ref. [55]. The overall error in the determination of the $fO_2$ in each experiment is estimated to be ~0.2 log units[53]. For experimental runs conducted under nominally dry conditions (no fluid added), we assumed an $aH_2O$ of 0.1 because such experiments are not strictly water-free for two reasons: (1) it is nearly impossible to avoid adsorbed water on the surface of the glass grains, and (2) hydrogen can be present in the pressure medium (gas) and may diffuse through the noble metal capsules. Thus in nominally dry experiments the silicate melts contained small amounts of water mainly present as OH groups (~0.3–1.0 wt.% depending on pressure and the extent of crystallization[56]). In nominally dry experiments, the oxygen fugacity was estimated at ~FMQ + 0.5.

Other experiments performed at reduced conditions were conducted in an IHPV pressurized with a mixture of Ar and $H_2$ gases (the maximum $H_2$ pressure given in the IHPV before heating was 7.5 bar). Hydrogen diffuses through the noble metal inside the capsules. If water is present in the experimental charge, the oxygen fugacity is controlled by the equilibrium reaction for water formation ($H_2 + O_2 = H_2O$). As a result, at a given $fH_2$, the $fO_2$ decreases with decreasing water activity in the experimental charge. The calculation of $fO_2$ is based on the equation

of ref. [57] [for further details see ref. [53]]. The $fH_2$ prevailing in the IHPV at high P and T was controlled with a Shaw membrane[50]. Various oxygen fugacities were obtained by varying the proportions of $H_2$ and Ar in the pressure medium. We estimate that the overall error in the calculated $fO_2$ is about 0.2 log units.

**Imaging and chemical analyses**. Back-scattered electron images of the experimental run products were acquired on the QEMSCAN FEI Quanta 650F at RWTH Aachen (Germany). Chemical analyses were performed using a CAMECA SX100 electron probe microanalyser at the University of Hannover (Germany). Analyses were performed with an accelerating voltage of 15 kV. For silicate glasses, we used a beam current of 8 nA and a defocused beam of 10–20 μm. For sulfide liquids, we used a beam current of 15 nA and a defocused beam of 2–20 μm. Mineral analyses were performed with a beam current of 15 nA and a focused beam (1 μm). For minerals, the counting times were 15–20 s on peak for each element. The peak counting times for glasses were 10 s for Si, Ti, Al, Fe, Mn, Mg, Ca, and S, and 8 s for alkalies. The elements Na, K, Si, Ca, and Fe were measured first. Subsequent analyses of F were performed using a second set of analytical conditions (60 nA), with counting times of 120 s on peak and 60 s for background[58]. For glasses and minerals, we used the following standards for Kα X-ray line calibration: albite for Na and Al, orthoclase for K, wollastonite for Si and Ca, $TiO_2$ for Ti, $Fe_2O_3$ for Fe, MgO for Mg, $Mn_3O_4$ for Mn, and $CaSO_4$ for $SO_3$, $CaF_2$ for F. Raw data were corrected using the PAP routine. The precision for oxide concentrations was better than 1%. No significant alkali loss (within uncertainty) was detected during measurements.

**Water determination**. Raman spectra were recorded with a Jobin-Yvon LabRam HR800 spectrometer (grating: 2400 gr/mm), equipped with an Olympus optical microscope and a long-working-distance LMPlanFI 100×/0.80 objective at GFZ Potsdam, Germany. We used a 488 nm excitation of a Coherent Ar + laser Model Innova 70C, a power of 180 mW (about 30 mW on sample), at a resolution <0.6 cm$^{-1}$. If necessary, the laser power was reduced by using density filters. Each unpolarized spectrum represents the accumulation of ten to twenty acquisitions of 10 seconds each. Spectra were collected at a constant laboratory temperature (20 °C) with a Peltier-cooled CCD detector, and the positions of the Raman bands were controlled and eventually corrected using the principal plasma lines in the Argon laser. The recommended and measured positions of the plasma lines in the fingerprint spectral region are not larger than 0.6 cm$^{-1}$. Water concentrations of the glasses were determined by confocal Raman spectroscopy following a standard method previously described in the literature[59,60]. A fresh, polished synthetic glass with a total of 8.06 wt.% $H_2O_T$, determined by Karl Fischer titration, was used as a reference standard. The composition of this standard was cross-checked using about 30 different glasses of basaltic to granitic bulk composition. The standard glass has been polished before each series of measurements. At high water concentration the differences in the bulk composition disappear, and the $H_2O$–OH-Raman band is dominant. Because the integral intensity of the $H_2O$–OH stretching band between about 2800 and 4000 cm$^{-1}$ increases directly, proportionally and linearly with the total water content it results a simple procedure for quantification, $I = 608 + 9219.15 \cdot H_2O_T$ ($r^2 = 0.9997$), in which $I$ is measured integral intensity (a.u.). The uncertainty on the water concentration is given in Supplementary Data 1.

**Data availability**. The authors declare that all relevant data are available within the article and its supplementary information files.

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

## Acknowledgements

Stefan Linsler, Julian Feige, and Chao Zhang are thanked for their help during experiments, sample preparation, and microprobe analyses. Don Lindsley is thanked for his kind help with the identification of oxide minerals. T.H. acknowledges support by the China Nature Foundation of Sciences (41502052 and 2016YFC0600502), a Marie Curie Individual Fellowship within the Horizon 2020—Research and Innovation Framework

Programme (656923), and the "Fundamental Research Funds for the Central Universities (2652015054)". B.C. is a Research Associate of the Belgian Fund for Scientific Research-FNRS. O.N. was supported by an Emmy Noether grant (DFG NA1171/1-1). This work was partly supported by DFG grant KO1723/20-1 and Chinese 973 program (2012CB416806).

## Author contributions

T.H., B.C., and O.N. designed the project, conducted the analyses, and led the writing of the manuscript. T.H. and O.N. conducted the experiments. R.T. analyzed the water concentrations in glasses. F.H., I.V., and Z.Z. contributed to the interpretation and did comprehensive editing.

## Additional information

**Competing interests:** The authors declare no competing interests.

