## [Peer Review File(PDF 397 kb) · Nature Communications]

Reviewers' comments:

Reviewer #1 (Remarks to the Author):

The manuscript by Hou et al is an exciting contribution to the origin of magnetite-apatite rocks, experimentally confirming that they can form from the crystallization of Fe-Ca-P-rich melts – the major goal of the study is that the authors have been able to expand the known immiscibility field of Fe-Ca-P melts by increasing the aH₂O and the redox state. Thus, the study adds significant information on how the magnetite-apatite deposits are formed, supporting that they can be of magmatic origin. The experiments also confirm many features that have been recently observed by different authors such as the existence of immiscible Fe-P melts at El Laco (Naslund et al 2002, 2003, Mungall et al., 2009, 2014), the existence of melt inclusions with similar compositions to those described here (Clark and Kontak, 2004; Naslund et al., 2009; Kamenetsky et al 2013, , Velasco et al., 2016), the local role of fluorine, the high redox state and water content of those melts and the formation of these systems by crustal contamination of primitive melts (Tornos et al., 2016, 2017).

Despite its major interest, my feeling is that the authors overemphasize the novelty of their study, make significant errors when confusing magnetite-apatite deposits with IOCG's and magmatic Ni-Cu deposits or don't give credits when quoting other people's work. In fact, in the abstract they include issues that are not discussed in their manuscript such as the validity of the models proposing an hydrothermal genesis. Also, it is not true that "Fe-Ca-P-rich and Si-poor melts have never been observed in natural or synthetic magmatic systems". There is abundant literature on the existence of such Fe-rich melts that are recorded in melt inclusions or in the magnetite-apatite deposits (references above) and Kamenetsky et al (2013) have observed an immiscibility gap identical only slightly smaller than that shown in the figure 4 (D_{Si} values of 0.2 instead 0.0 as in this study). Also, the authors do not present any data that show that assimilation of sediments (by silicate melts?) can produce such Fe-Ca-P-rich magmas. They just highlight an already published statement without adding new data. Also, the authors do not discuss the results of Lester et al (2012, 2013) that seem to reach similar conclusions.

Line 33. Discordant? Some of the major magnetite-apatite deposits are stratabound such as Kiruna or El Laco

Line 41. Unconsistent with what said above

Line 43. Reference?

Line 45. Chen et al (2019) and Broughm et al (2017) say very little about this. There are papers that discuss this problem in much more detail (e.g., Velasco et al., 2016)

Line 48. The inclusions described by these authors are in ortho- and clinopyroxene but also (and dominantly) in plagioclase.

Lines 51-52. There is abundant literature that has suggested this in geologic systems and there is melt inclusion evidence that these immiscible melts coexist. However, Hou et al are, to my knowledge, the first ones obtaining such extreme endmembers.

Line 61 and onwards. The authors do not even quote the presence of actinolite/diopside, a common component of these rocks.

Line 78. Pyrite is not so common in IOA, which are characterized by the almost lack of sulphides. Pyrite is always late in this mineralization, postdating and replacing magnetite.

Line 82. In your experiments, you obtain fayalite, which is a very uncommon mineral in magnetite-apatite deposits. In fact, and to my knowledge, it has never been recorded except in the unusual fluorite Vergenoeg deposit – which lacks of apatite.

Line 84 and 104. Ilmeno-magnetite is another of your products while, as the authors quote above, magnetite-apatite deposits are characterized by the lack of Ti-rich magnetite.

Line 99. This statement is really interesting. Droplets of an immiscible sulphide-rich liquid... and magnetite apatite deposits lack of sulphides coeval with magnetite. In line 131 you quote it as close to stoichiometric FeS (pyrrhotite?). In line 78 you quote pyrite as the stable phase.

Line 113. The authors quote that there is somewhat abundant MgO in the immiscible Fe-rich melt. Why they call them Fe-Ca-P melts and not Fe-P-Mg-Ca-Ti melts?

Line 150. This is not true. Your experiments quote abundant MgO and TiO₂ also.

Line 173. These are the conclusions of the previous papers of Tornos et al (2017).

Line 185. Do you mean that all the dacite-rhyolite found near magnetite-apatite deposits is the immiscible silica-rich melt? In some districts (i.e., Kiruna) the volume of these rocks is enormous. Conversely, rhyolite and dacite are lacking in other districts (Andes).

Line 186. Here there is a major confusion with tectonic environments and magma geochemistry. IOA deposits are only located in convergent margins in the Andes and in the lower Yangtze. In other settings they are in extensional intraplate settings (Hitzman et al., 1992; Williams et al., 2005). The genesis of hydrous melts is not only related to slab subduction but also melting of crustal rocks.

Line 192. Which kind of isotopic studies? The work of Barton et al is on IOCG deposits and not magnetite-apatite and proposes the involvement of basinal brines, not melts, in the genesis of these deposits. They just provide isotopic evidence that the sulphur derives from the evaporites but that does not mean that the rocks are crustal. This needs a much more rigorous approach.

Line 194 and onward. This is the model by the recently published papers of Turnos et al (2017a and b).

Line 199. Panzihua is not a magnetite-apatite deposit. It is a Ni-Cu deposit with magnetite intergrown with sulphides. As in many other Ni-Cu deposits, crustal contamination is the key for forming an immiscible sulphide-rich melt. Anhydrite has been systematically recorded in true magnetite-apatite deposits (Kiruna, El Laco).

Table 1. Some of the melts are so P-rich that cannot be called Fe-P melts. They have more CaO and MgO than Fe! Conclusions similar to the experiments of Mungall et al (2009, 2014). See also the description of such melts by Naslund et al 2002, 2003).

Reviewer #2 (Remarks to the Author):

Review of NCOMMS-17-28343-T

This is a very nice piece of work, which answers a long standing question in Earth Sciences and basically provides a recipe for the genesis of iron oxide – apatite ore deposits. The data is convincing overall, and the interpretations seem to be fully supported by the data. The presentation is of high quality throughout, and the manuscript reads very well. I strongly support publication in Nature Communications pending on some minor revisions based on my comments below:

Line 141-143: Sulfur is siderophile only in 2- oxidation state, which is not the case here. For oxidized sulfur, I would rather tie the difference to the degree of melt polymerization (e.g. see Zajacz 2015, GCA).

Line 174-178: This section seems very simplified and somewhat speculative. For example, 2.6 wt% F cannot serve as a ligand for about 20 wt% Ca.

Line 184-185: Does this statement also hold for FeO concentration?

Figure 4: On this diagram, the immiscibility field closes at $DSi=0.87$ instead of $DSi=1$. This must be a mistake, DSi should equal 1 at the critical point by definition.

Methods:

Line 31-32: It needs to be explained how the intrinsic fO_2 of IHPV was defined at FMQ+3.3 at water saturation. If $fH_2=0$ in pure argon pressure medium, this should yield much high fO_2 at $aH_2O=1$ (i.e. as constrained by the equilibrium constant of the water dissociation reaction at the run P, T). No satisfactory explanation is given in the Supplementary Material either, as opposed to the statement in line 36-37. I think it is very important to substantiate this, because fO_2 is a critically important variable in this study.

Line 70-73: The quantification of the water concentration from the Raman spectra needs more detailed explanation. Most commonly, the water band area is normalized by the area of one of the major silicate bands for this purpose, but this method generally requires matrix-matched standardization. Therefore, much more specifics are needed on the standards used as well. For example, I don't see how one could use the same standard for the immiscible melts in this study with contrasting compositions. What is the uncertainty on the water concentration determination?

Supplementary Material:

Supplementary Figure 1: Why only plot liquid fields based on literature data? Any reason why the composition of the coexisting liquids from this study cannot be plotted on this diagram for comparison?

Oxygen fugacity calculations section:

As pointed out above, the determination of intrinsic fO_2 needs more explanation. Also, further down, would 0.3-1.0 wt% dissolved water really correspond to $a_{H_2O}=0.01$?

The language of the last paragraph could use some improvement.

Zoltan Zajacz

17/12/2017

Thank you very much for your comments and kind help. All comments from the two reviewers are very constructive and helpful. In the following paragraphs, we answer the questions and address point by point the issues pointed out by reviewers.

Before presenting our replies to reviewers, we first want to emphasize that our study is an experimental approach that aims at understanding phase equilibria in magmatic systems related to Iron-Oxide-Apatite (IOA) ores with broad implications for these deposits globally. In other words, our study does not focus on any specific case, and for example on El Laco to which Reviewer #1 continuously refers to. We evaluate the formation of IOA ores on a large-scale perspective rather than on the scale of individual rocks or deposits. The ore-forming processes we propose should not be evaluated from a regional perspective, but from phase equilibria and crystallization conditions as evidenced by our new experiments.

Replies to Reviewer #1,

The manuscript by Hou et al is an exciting contribution to the origin of magnetite-apatite rocks, experimentally confirming that they can form from the crystallization of Fe-Ca-P-rich melts – the major goal of the study is that the authors have been able to expand the known immiscibility field of Fe-Ca-P melts by increasing the a_{H_2O} and the redox state. Thus, the study adds significant information on how the magnetite-apatite deposits are formed, supporting that they can be of magmatic origin. The experiments also confirm many features that have been recently observed by different authors such as the existence of immiscible Fe-P melts at El Laco (Naslund et al 2002, 2003, Mungall et al., 2009, 2014), the existence of melt inclusions with similar compositions to those described here (Clark and Kontak, 2004; Naslund et al., 2009;

Kamenetsky et al 2013, Velasco et al., 2016), the local role of fluorine, the high redox state and water content of those melts and the formation of these systems by crustal contamination of primitive melts (Tornos et al., 2016, 2017).

Thank you very much for the compliment!

- 1. Despite its major interest, my feeling is that the authors overemphasize the novelty of their study, make significant errors when confusing magnetite-apatite deposits with IOCG's and magmatic Ni-Cu deposits or don't give credits when quoting other people's work.*

First of all, we have added all the references mentioned by Reviewer 1 into the text in order to give credits to our colleagues from the IOA community. Please see the revised version of the manuscript.

Secondly, although sulfide is always late in this type of mineralization and locally replacing magnetite, it is likely that sulfur got involved initially in the primordial mineralizing-process. Thus, in order to simulate the role of sulfur during magmatic processes, we deliberately added minor sulfur concentrations in three runs. Since all of our experiments are free of Ni and Cu, there is no possible confusion between IOA-IOCG and Ni-Cu deposits, these last ones being not mentioned at all in our manuscript. It should however be noted that IOCG deposits have considerable amounts of sulfide and IOA deposits are commonly regarded as an end-member of IOCG series (Hitzmann et al., 1992).

- 2. In fact, in the abstract they include issues that are not discussed in their manuscript such as the validity of the models proposing an hydrothermal genesis.*

In this study we are aiming to elucidate the primordial process for the formation of IOA deposits. In order to do so, we conducted experiments simulating magmatic processes. We however find reasonable to include the debate on magmatic vs. hydrothermal genesis in the introduction to put the study into a broader context and to make the reader aware that other processes than magma crystallization have been proposed to explain the genesis of IOA deposits. Our experiments, conducted to constrain the phase equilibria at magmatic temperature, constrain the primordial role of magmas for the enrichment in iron and phosphorous. However, we totally agree that hydrothermal process played an important role in the lower-temperature redistribution of elements but we agree that these late-stage processes did not lead to the formation of deposits (e.g., Knipping et al., 2015; Tornos et al., 2016, 2017).

3. *Also, it is not true that “Fe-Ca-P-rich and Si-poor melts have never been observed in natural or synthetic magmatic systems”. There is abundant literature on the existence of such Fe-rich melts that are recorded in melt inclusions or in the magnetite-apatite deposits (references above) and Kamenetsky et al (2013) have observed an immiscibility gap identical only slightly smaller than that shown in the figure 4 (DSi values of 0.2 instead 0.0 as in this study).*

Yes, we know the reference and this study is thoroughly discussed in our paper (actually, B. Charlier is co-author in both studies). It is true that Kamenetsky et al. (2013) have observed an immiscibility an important gap but it is not as large as the one reported in our paper. Moreover, the melt inclusions reported in Kamenetsky et al. (2013) are hosted in native iron, which implies that they equilibrated at very reducing conditions close to the iron-wustite (IW) solid buffer. Such reducing environment is not consistent with the formation of IOA deposits which usually record relatively

oxidizing conditions above QFM buffer (e.g., Lester et al. 2013; Tornos et al., 2016, 2017). This is extremely important for iron speciation and the nature of liquidus phases crystallizing from the magmas. This difference regarding oxygen fugacity is an important part of the motivation to conduct new experiments, since the contrasted melt compositions observed by Kamenetsky et al. (2013) are not suitable to interpret the genesis of IOA deposits.

4. Also, the authors do not present any data that show that assimilation of sediments (by silicate melts?) can produce such Fe-Ca-P-rich magmas. They just highlight an already published statement without adding new data.

First of all, we have added the reference for that statement into the text of our original submission (Tornos et al. 2017). One should keep in mind that this paper is an experimental study aiming to test the model of ‘oxide melt’ for IOA ores, which had been previously proposed in the literature (e.g., Philpotts, 1967; Frietsch 1978; Nyström and Henriquez 1994; Naslund et al. 2002). We evaluate the formation of IOA ore on a large-scale perspective rather than on the scale of individual rocks or deposits. Hence it makes perfect sense to give global implications for IOA deposits and to test existing hypotheses, without presenting new geochemical analysis of natural samples for individual localities.

Specifically, based on experiments under relevant conditions, one of our major new findings is that elevation of oxygen fugacity could produce Fe-Ca-P enriched melts by immiscibility. Thus any processes occurring in natural systems that could oxidize the ore-forming magmas has the potential to form such melts, including contamination of crustal rocks, e.g. carbonates or evaporates. The O-Sr-Nd isotopes

of both igneous rocks and IOA ores are supportive for this inference as discussed in our manuscript (Tornos et al., 2016).

5. *Also, the authors do not discuss the results of Lester et al (2012, 2013) that seem to reach similar conclusions.*

The results of Lester et al. (2013) have now been taken into consideration and discussed at the end of *Introduction* section: “For example, in the simple system of SiO₂-FeO-Al₂O₃-K₂O with presence of volatile, Lester et al. (2013) produced Fe-rich melts with SiO₂ content of >20 wt% by immiscibility, which is still not a silica-free oxide melt.”

It is worth noting that this clearly means their conclusions are significantly different to ours with respect to the formation of IOA deposits.

6. *Line 33. Discordant? Some of the major magnetite-apatite deposits are stratabound such as Kiruna or El Laco*

We have deleted “discordant” from that sentence.

7. *Line 41. Unconsistent with what said above*

Here we present the current debate for the formation of IOA deposits, i.e. hydrothermal vs. magmatic models. Accordingly, we first described the hydrothermal model, and then followed by those for the magmatic model. Thus it is logically correct and the two parts do not mutually agree with each other as expected.

8. *Line 43. Reference?*

We have added four references here, Naslund et al. (2002), Chen et al. (2010) and Tornos et al. (2016 and 2017).

9. *Line 45. Chen et al (2010) and Broughm et al (2017) say very little about this. There are papers that discuss this problem in much more detail (e.g., Velasco et al., 2016)*

We have replaced these two references by Velasco et al. (2016).

10. *Line 48. The inclusions described by these authors are in ortho- and clinopyroxene but also (and dominantly) in plagioclase.*

We changed this sentence into “The development of immiscibility is supported by the coexistence of two types of melts in glassy matrices and inclusions hosted by phenocrysts in the ore and in andesitic wall rocks.”

11. *Lines 51-52. There is abundant literature that has suggested this in geologic systems and there is melt inclusion evidence that these immiscible melts coexist. However, Hou et al are, to my knowledge, the first ones obtaining such extreme end-members.*

Yes, we combined some of the representative compositions of these immiscible melts coexist as illustrated in Figs. 2 and 4. However, as we stated clearly in the text, the Fe-rich melt inclusions still contain considerable amount of silica and they are not compositionally similar to the high grade, i.e., almost SiO₂-free, IOA ores. This was our motivation to conduct this study. In contrast to the melts observed in the inclusions, the Fe-rich immiscible melts that we produced experimentally in this study are almost SiO₂-free which and are capable of crystallizing the high-grade IOA ores. To the best of our knowledge, similar melts had not been observed either in natural melt inclusions or in experiments. Even in Lester et al. (2013) such a melts were not

reported although the authors of this study used a simple system which is more favorable to extreme, but not geologically relevant compositions.

In summary, we present the first results that prove experimentally that almost SiO₂-free melts can be produced by immiscibility under geological environment relevant for IOA deposits in a multi-component, geologically-realistic system. As pointed out by Reviewer #1, we are “*the first ones obtaining such extreme end-members*”, which endorses the novelty of our study.

12. Line 61 and onwards. The authors do not even quote the presence of actinolite/diopside, a common component of these rocks.

We now clearly quote the presence of actinolite/diopside in the revised version, as stated below,

“Extreme enrichment of apatite and iron oxide over silicate minerals, as observed in IOA deposits, cannot simply result from differential crystal settling in an iron-rich silicate melt. This is because, with the exception of plagioclase, common silicate minerals (actinolite and diopside) are denser than the melt and would sink along with the oxides”

13. Line 78. Pyrite is not so common in IOA, which are characterized by the almost lack of sulphides. Pyrite is always late in this mineralization, postdating and replacing magnetite.

Yes, this is true. However, as mentioned by Reviewer 1, some IOA deposits contain sulfides. In order to thoroughly investigate and understand the general mechanisms by which IOA deposits formed, it is necessary to take into account that they contain (even only a little bit) sulfur. This is why we added sulfur in the

experiments simulating magmatic stage. Although sulfides appear as being late phases in the crystallization sequence, there is still a possibility that these sulfur played a role during the primordial process and were re-worked during post-magmatic hydrothermal processes. We however strongly emphasize that the presence of sulfur in our experiments has no implications for the compositional range of immiscible melts that we produced. Adding sulfur was therefore important to understand how sulfides form in IOCG deposits but does not have implication for the development of immiscibility and the formation of apatite and oxide ore.

14. Line 82. In your experiments, you obtain fayalite, which is a very uncommon mineral in magnetite-apatite deposits. In fact, and to my knowledge, it has never been recorded except in the unusual fluorite Vergenoeg deposit – which lacks of apatite.

The experiment conducted under fO_2 corresponding to close to QFM commonly crystallize quartz (tridymite) + fayalite + magnetite simultaneously as expected. The occurrence of fayalite in IOA and IOCG deposits are not so uncommon as addressed by Reviewer #1. For example, fayalite had been found both spatially and temporally associated with IOA deposit in Sri Lanka (He et al., 2017), in Lyon Mountain, Adirondack Mountains, New York State (Valley et al., 2011), St Francois Mountains, Missouri (Kisvarsanyi and Kisvarsanyi, 1990). Moreover, fayalite is a common mineral phase in IOCG deposits (Porter et al., 2010), for example, Carajas district in Brazil (Xavier et al., 2008).

15. Line 84 and 104. Ilmeno-magnetite is another of your products while, as the authors quote above, magnetite-apatite deposits are characterized by the lack of Ti-rich magnetite.

As discussed by Velasco et al. (2016), low-Ti magnetite in IOA deposits does not necessarily mean hydrothermal in origin, because considerable re-equilibration between the magnetite and the parental melt may occur during the onset or during the oxy-exsolution at shallow depths prior to magma ascent. This seems to be the case for the magnetite in IOA deposits which usually shows significant variations in the highly compatible transition elements such as Ti, V, Cr and Ni (e.g., Knipping et al., 2015). It is therefore reasonable to suggest that all or part of the low-Ti magnetite crystals observed in IOA/IOCG deposits initially had a much higher Ti content when they formed at the magmatic stage. Our experiments of course only record the magmatic stage and not the late-stage processes during which Ti may be lost from the magnetite lattice. In order to simulate the potentially high-Ti feature during the magmatic stage, we may have added excessive Ti in the starting mixtures. However, there's nothing necessarily wrong with such a methodology. For example, in the Se-Chahun IOA deposit, Bafq district, Iran (Bonyadi et al., 2011), magnetite crystals are always featured with Ti-rich core (~2.5 wt.%) and ilmenite exsolution is quite common in magnetite, which is notable, as being of an indicator for high-Ti bulk composition of primary oxide minerals.

16. Line 99. This statement is really interesting. Droplets of an immiscible sulphide-rich liquid... and magnetite apatite deposits lack of sulphides coeval with magnetite. In line 131 you quote it as close to stoichiometric FeS (pyrrhotite?). In line 78 you quote pyrite as the stable phase.

Please see reply #13 for the question “magnetite apatite deposits lack of sulphides coeval with magnetite.” Additionally, we had already replaced ‘pyrite’ in line 78 with pyrrhotite as it is close to stoichiometric FeS.

17. Line 113. The authors quote that there is somewhat abundant MgO in the immiscible Fe-rich melt. Why they call them Fe-Ca-P melts and not Fe-P-Mg-Ca-Ti melts?

First of all, these melts are low in Ti, i.e. as low as < 1wt.% TiO₂ (please see row 5 in Table 1). This is consistent with the relatively low Ti signature of the IOA magnetites. In ‘Fe-Ca-P’ melts we defined, the MgO content (up to 7.5 wt.%) is the highest among other elements except FeO_{tot}, CaO and P₂O₅, which is still two times below the lowest content of FeO_{tot} (18.88 wt.%). Hence, FeO_{tot}, combined with CaO (~27wt.%) and P₂O₅ (~26wt.%), are the major constituents in the ‘Fe-Ca-P’ melts. Moreover, except MgO, these melts also contain ~5 wt.% Al₂O₃, which is not really minor components. Nevertheless, considering the major mineral assemblage in the iron oxide apatite ores, i.e. magnetite + apatite, we believe the name (Fe-Ca-P) better reflect the characteristics of such an immiscible Fe-rich melt. Please also see reply #25 for the more discussion for the names of the immiscible Fe-rich melts.

18. Line 150. This is not true. Your experiments quote abundant MgO and TiO₂ also.

Thank you very much for the suggestion, we changed the sentence into “With cooling, immiscible melt pairs become increasingly contrasted in composition, but dry Fe-rich melts reported so far have never approached the extreme, Si-free, composition of IOA ores (predominantly iron, calcium, phosphorous, with minor magnesium and titanium)”. Minor magnesium and titanium may be incorporated into magnetite and exsolution of ilmenite in some cases. Moreover, except magnetite and apatite, our Fe-Ca-P melts may inevitably crystallize some phases containing all the other elements (e.g., MgO, MnO, Al₂O₃ and SiO₂ etc.), for example, diopside,

actinolite, amphibole, epidote, titanite and allanite. These phases are commonly seen in IOA deposits but may be interpreted as secondary phases formed by extensive alteration.

19. Line 173. These are the conclusions of the previous papers of Tornos et al (2017).

We now cite Tornos et al. (2016 and 2017) in the revised version.

Accordingly, we changed the sentence into “Thus, increasingly hydrous and oxidizing conditions can explain the formation of Fe-P-enriched silicate liquids (Fig. 4), which is consistent with the recent studies on El Laco deposit (Tornos et al., 2016; 2017)”.

20. Line 185. Do you mean that all the dacite-rhyolite found near magnetite-apatite deposits is the immiscible silica-rich melt? In some districts (i.e., Kiruna) the volume of these rocks is enormous. Conversely, rhyolite and dacite are lacking in other districts (Andes).

1) Si-rich rocks, including rhyolite and dacite are widely recognized in IOA deposits, even in the districts like Andes. For example, dacitic rocks are well exposed in Marcona in Peru (Chen et al., 2010), and a recent study on El Laco had also revealed that a high-K subalkaline rhyodacitic melt is present in the groundmass of the andesite and in most melt inclusions (Tornos et al., 2017).

2) During magmatic evolution, once the liquid line of descent (LLD) crossed the two-liquid field, immiscibility that split the homogenous melt into Fe-rich and Si-rich conjugates will develop. The compositions of immiscible pairs define a locus between which immiscibility develops. Any composition that plots on the

mixing trend between the equilibrium immiscible pairs would unmix, the proportion of the two liquids being defined by the lever rule. If the liquid line of descent left the two liquid field by simple fractional crystallization and compositional evolution of the bulk liquid, the most Si-rich, i.e., evolved liquids can be produced. Thus, the Si-rich magmas could be only immiscible Si-rich melts or a combination of immiscible Si-rich melts and most evolved magmas after LLD out of the solvus. Hence, relative volume of evolved melt and ore material could be varied significantly, i.e., large volumes of dacite/rhyolite in some places and very small volumes in other places with a single process.

21. Line 186. Here there is a major confusion with tectonic environments and magma geochemistry. IOA deposits are only located in convergent margins in the Andes and in the lower Yangtze. In other settings they are in extensional intraplate settings (Hitzman et al., 1992; Williams et al., 2005). The genesis of hydrous melts is not only related to slab subduction but also melting of crustal rocks.

Thank you very much for pointing this out, we added “melting of crustal rocks” into the statement, “First, IOA deposits are commonly located in convergent settings where slab dehydration leads to the formation of water-bearing magmas⁴⁴. There are also commonly observed in extensional intraplate setting where crustal rocks melting (Hitzman et al., 1992; Williams et al., 2005)^{*45-46} may also produce hydrous magmas¹⁰.”

****Note: the format of citation will be kept consistent as superscript numbers in the text.***

22. Line 192. Which kind of isotopic studies? The work of Barton et al is on IOCG deposits and not magnetite-apatite and proposes the involvement of basinal

brines, not melts, in the genesis of these deposits. They just provide isotopic evidence that the sulphur derives from the evaporites but that does not mean that the rocks are crustal. This needs a much more rigorous approach.

Recent papers published by Tornos and his group had already provided a rigorous approach in a support for our suggestion which is mainly based on experimental results. For example, in the case study of El Laco, Tornos et al. (2016 and 2017) presented isotope data from the host andesite ($^{87}\text{Sr}/^{86}\text{Sr}$: 0.7066–0.7074; ϵNd : –5.5 to –4.1; $\delta^{18}\text{O}_{\text{whole rock}}$: 7.2–9.6‰; $\delta^{18}\text{O}_{\text{magnetite}}$: 5.1–6.2‰) and an underlying andesite porphyry ($^{87}\text{Sr}/^{86}\text{Sr}$: 0.7075–0.7082; ϵNd : –5.9 to –4.6). These value reflect the interaction of a primitive mantle melt with Andean crustal rocks. It worthy noting that a rock with evaporate can hardly be anything else than crustal.

Since we had already cited Tornos et al. (2017) in this sentence, we thus just changed it into “Indeed, systematic O-Sr-Nd isotopic studies of IOA ores suggest a significant crustal component⁴⁸”.

23. Line 194 and onward. This is the model by the recently published papers of Tornos et al (2017a and b).

We checked the publications from Tornos and his group. We could only find one paper published on *Economic Geology* in 2017. Reviewer 1 is presumably referring to the paper published in *Geology* in 2016 (Tornos et al., 2016). Since we had cited Tornos et al. (2017) here in the last version, and in this revised version, we simply added the another reference to Tornos et al. (2016).

24. Line 199. Panzhihua is not a magnetite-apatite deposit. It is a Ni-Cu deposit with magnetite intergrown with sulphides. As in many other Ni-Cu deposits, crustal contamination is the key for forming an immiscible sulphide-rich melt.

Anhydrite has been systematically recorded in true magnetite-apatite deposits (Kiruna, El Laco).

We deleted the reference to the Panzhihua deposit and cited Tornos et al. (2017), in which presence of anhydrite in El Laco had been reported.

25. *Table 1. Some of the melts are so P-rich that cannot be called Fe-P melts.*

They have more CaO and MgO than Fe!

In order to make this issue clear and explain why we defined Fe-P and Fe-Ca-P melts, we present the composition of the three types of Fe-rich immiscible melts produced experimentally here,

	SiO ₂	TiO ₂	Al ₂ O ₃	FeO _{tot}	MnO	MgO	CaO	Na ₂ O	K ₂ O	P ₂ O ₅	F
Fe-rich silicate melts	22.69-32.66	0.81-2.82	2.44-6.14	33.85-40.84	0.99-2.73	1.48-4.21	6.02-8.49	0.57-0.83	0.25-0.69	8.55-18.07	0.23-0.41
Fe-P melts	3.09-4.55	0.99-1.53	2.18-3.90	32.63-36.36	2.99-3.28	4.87-5.22	5.13-5.74	1.74-1.90	0.77-1.00	39.02-39.40	0.48-0.60
Fe-Ca-P melts	3.33-3.40	0.98	5.19-5.72	18.88-19.15	3.53-5.66	7.23-7.48	27.16-27.20	2.19-2.50	0.65-0.69	25.71-26.43	2.51-2.71

- 1) If you look at this table carefully, you will see that no Fe-rich melt has more MgO than FeO_{tot}.
- 2) In '**Fe-Ca-P**' melts we defined, the MgO content (up to 7.5 wt.%) is the highest among other elements except Fe-Ca-P, which is still two times below the lowest content of FeO_{tot} (18.88 wt.%). Hence, FeO, combined with CaO (~27wt.%) and P₂O₅ (~26wt.%), are the major constituents in the '**Fe-Ca-P**' melts.
- 3) In the '**Fe-P**' melts, obviously FeO (33-36 wt.%) and P₂O₅ (~39 wt.%) are the two major components, any other elements are subordinate, for example, both MgO and CaO contents are up to 5-6 wt.%, which is about seven times below the lowest content of FeO. This why we call such melts '**Fe-P**' melts.

26. *Conclusions similar to the experiments of Mungall et al (2009, 2014). See also the description of such melts by Naslund et al 2002, 2003).*

We cite Mungall et al. (2018) in which his systematic experiments are presented and the two papers of Naslund in the revised version.

Naslund et al. (2002) described FeS liquids and Fe-S-O liquids that extends from pure FeS liquids to liquids with O/S > 1.3 (Larocque et al., 2000; Rose and Brenan, 2001). They are immiscible with silicate melts, and are characterized by high Fe and low Ti contents. With these melts, it would be possible to explain the formation of oxide (P-poor) ores in El Laco if sulfur was lost during eruption or oxidation subaerially. They however do not explain the formation of apatite-bearing deposits. In contrast to the studies described above, our study proposes that Fe-Ca-P melts are immiscible with silicate melts under oxidizing condition and in presence of volatile. We believe that these melts and our model have global implications for the formation of IOA deposits and not only for El Laco, which is known as a peculiar deposit compared to most others.

Replies to Reviewer #2, Dr. Zoltan Zajacz,

27. *Line 141-143: Sulfur is siderophile only in 2- oxidation state, which is not the case here. For oxidized sulfur, I would rather tie the difference to the degree of melt polymerization (e.g. see Zajacz 2015, GCA).*

We changed the sentence into “Due to the differences of the degree of melt polymerization in the immiscible conjugates(Zajacz, 2015)⁴⁰, SO₃ partitions preferentially into the Fe-rich melts (Fe-rich melt: 1.07–1.53 wt.% SO₃; D_{SO_3} : 21.5-

38.3) and sulfur concentrations in the Si-rich immiscible melts are therefore very low (0.03–0.06 wt.%).”

28. Line 174-178: This section seems very simplified and somewhat speculative.

For example, 2.6 wt% F cannot serve as a ligand for about 20 wt% Ca.

We had never said that 2.6 wt.% F serves as a ligand for 20 wt.% Ca. We were trying to express that in melts, Ca complexes with F. This will effectively decrease Ca activity in the silicate melt before immiscibility developed, but that does not mean that the melt does not contain any Ca. Therefore, this is not speculative, and it is true that if Ca complexes with F, Ca activity will be decreased. Besides, we know that small variations in melt compositions or activities of elements could lead to immiscibility.

29. Line 184-185: Does this statement also hold for FeO concentration?

Yes, it does. Table 1 shows that the Si-rich conjugates of Fe-P melts contain 2.48-3.20 wt.% FeO_{tot}, and those of Fe-Ca-P melts have 2.99-3.05 wt.%. These melts resemble to typical rhyolite and dacite in term of FeO concentration.

30. Figure 4: On this diagram, the immiscibility field closes at DSi=0.87 instead of DSi=1. This must be a mistake, DSi should equal 1 at the critical point by definition.

That is true, we changed this diagram. DSi equals 1 at the critical point in the new version. Please see the updated Fig. 4.

Methods:

31. Line 31-32: It needs to be explained how the intrinsic fO_2 of IHPV was defined at FMQ+3.3 at water saturation. If $fH_2=0$ in pure argon pressure medium, this should yield much high fO_2 at $aH_2O=1$ (i.e. as constrained by the equilibrium constant of the water dissociation reaction at the run P, T). No satisfactory explanation is given in the Supplementary Material either, as opposed to the statement in line 36-37. I think it is very important to substantiate this, because fO_2 is a critically important variable in this study.

We provided a detailed explanation below and changed the description in the Supplementary Material accordingly. Please see the appendix for details.

32. Line 70-73: The quantification of the water concentration from the Raman spectra needs more detailed explanation. Most commonly, the water band area is normalized by the area of one of the major silicate bands for this purpose, but this method generally requires matrix-matched standardization. Therefore, much more specifics are needed on the standards used as well. For example, I don't see how one could use the same standard for the immiscible melts in this study with contrasting compositions. What is the uncertainty on the water concentration determination?

We did not normalize the water band area by silicate. We used a simpler but accurate method following which the intensity of the complex H_2O-OH band (where the integral intensity is directly proportional to the bulk water concentration) is directly determined. Accordingly, we added these sentences into the *Method* section to explain the measurement clearly,

“A fresh, polished synthetic glass with a total of 8.06 wt.% H_2O_T , determined by Karl Fischer titration, was used as a reference standard. The composition of this

standard was cross-checked using about 30 different glasses of basaltic to granitic bulk composition. the standard glass has been polished before each series of measurements. At high water concentration the differences in the bulk composition disappear, and the H₂O-OH-Raman band will be dominant. Because the integral intensity of the H₂O-OH stretching band between about 2800 and 4000cm⁻¹ increases directly, proportionally and linearly with the total water content it results a very simple procedure for quantification, $I = 608 + 9219.15 \cdot H_2O_T$ ($r^2=0.9997$), in which I is measured integral intensity (a.u.). The uncertainty on the water concentration is given in Supplementary Table 4".

Supplementary Materials

33. Supplementary Figure 1: Why only plot liquid fields based on literature data?

Any reason why the composition of the coexisting liquids from this study cannot be plotted on this diagram for comparison?

Supplementary Figure 1 was initially designed at showing the compositions of natural rocks that we statistically investigated to calculate the starting compositions of our experimental end-members. However, we see no problem to show the experimental results here as well. We have added our experimental data on Supplementary Figure 1.

34. As pointed out above, the determination of intrinsic fO_2 needs more explanation. Also, further down, would 0.3-1.0 wt% dissolved water really correspond to $a_{H_2O}=0.01$?

We re-calculated the aH₂O based on the model of Burnham (1994). aH₂O of 0.1 corresponds to 0.3-1.0 wt.% dissolved water. Please see the new section on oxygen fugacity in the *Supplementary material* for additional details.

35. *The language of the last paragraph could use some improvement.*

The manuscript has been checked and improved by an English native speaker.

References mentioned in our replies

- Bonyadi, Z., Davidson, G. J., Mehrabi, B., Meffre, S., Ghazban, F. (2011). Significance of apatite REE depletion and monazite inclusions in the brecciated Se–Chahun iron oxide–apatite deposit, Bafq district, Iran: insights from paragenesis and geochemistry. *Chemical Geology*, 281(3), 253-269.
- Burnham, C.W. (1994) Development of the Burnham model for prediction of H₂O solubility in magmas. In M.R. Carroll, and J.R. Holloway, Eds. Volatiles in Magmas, 30, p. 123–129. *Reviews in Mineralogy*, Mineralogical Society of America, Chantilly, Virginia.
- Chen H., Clark A.H., Kyser T.K. (2010). The Marcona magnetite deposit, Ica, South-Central Peru: A product of hydrous, iron oxide-rich liquids? *Economic Geology* 105, 1441–1456.
- Chung, H.Y., Mungall, J.E. (2009). Physical constraints on the migration of immiscible fluids through partially molten silicates, with special reference to magmatic sulfide ores. *Earth and Planetary Science Letters*, 286(1), 14-22.
- Frietsch, R. (1978). On the magmatic origin of iron ores of the Kiruna type. *Economic Geology*, 73(4), 478-485.
- He, X.F., Santosh, M., Tsunogae, T., Malaviarachchi, S.P. (2018). Magnetite-apatite deposit from Sri Lanka: Implications on Kiruna-type mineralization associated with ultramafic intrusion and mantle metasomatism. *American Mineralogist*, 103(1), 26-38.
- Hitzman, M.W., Oreskes, N., Einaudi, M.T. (1992). Geological characteristics and tectonic setting of Proterozoic iron oxide (Cu-U-Au-REE) deposits. *Precambrian Research*, 58(1-4), 241-287.
- Kamenetsky, V.S., Charlier, B., Zhitova, L., Sharygin, V., Davidson, P., Feig, S. (2013). Magma chamber–scale liquid immiscibility in the Siberian Traps represented by melt pools in native iron. *Geology*, 41(10), 1091-1094.
- Kisvarsanyi, E.B., Kisvarsanyi, G. (1990). Alkaline granite ring complexes and metallogeny in the Middle Proterozoic St. Francois terrane, southeastern Missouri, USA. Mid-Proterozoic Laurentia-Baltica: *Geological Association of Canada Special Paper*, 38, 433-446.
- Knipping, J.L., Bilenker, L.D., Simon, A.C., Reich, M., Barra, F., Deditius, A.P., Munizaga, R. (2015). Giant Kiruna-type deposits form by efficient flotation of magmatic magnetite suspensions. *Geology*, 43(7), 591-594.

- Larocque, A.C., Stimac, J.A., Keith, J.D., Huminicki, M.A. (2000). Evidence for open-system behavior in immiscible Fe–S–O liquids in silicate magmas: implications for contributions of metals and sulfur to ore-forming fluids. *The Canadian Mineralogist*, 38(5), 1233-1249.
- Lester, G.W., Clark, A.H., Kyser, T.K., Naslund, H.R. (2013). Experiments on liquid immiscibility in silicate liquids with H₂O, P, S, F and Cl: implications for natural magmas. *Contributions to Mineralogy and Petrology* 166, 329–349.
- Mungall, J. E., Brenan, J. M. (2014). Partitioning of platinum-group elements and Au between sulfide liquid and basalt and the origins of mantle-crust fractionation of the chalcophile elements. *Geochimica et Cosmochimica Acta*, 125, 265-289.
- Naslund, H.R., Henriquez, F., Nystroem, J.O., Vivallo, W., and Dobbs, F.M. (2002) Magmatic iron ores and associated mineralisation: Examples from the Chilean high Andes and coastal Cordillera, in Porter, T.M., ed., *Hydrothermal Iron Oxide Copper-Gold and Related Deposits: A Global Perspective*: Adelaide, Australia, PGC Publishing, 2, 207–226.
- Nyström, J.O., Henriquez, F. (1994). Magmatic features of iron ores of the Kiruna type in Chile and Sweden; ore textures and magnetite geochemistry. *Economic Geology*, 89(4), 820-839.
- Philpotts, A.R. (1967). Origin of certain iron-titanium oxide and apatite rocks. *Economic Geology*, 62(3), 303-315.
- Porter, T.M. (2010). Current understanding of iron oxide associated-alkali altered mineralised systems. Part 1—An overview. *Hydrothermal iron oxide copper-gold and related deposits: a global perspective*, 3, 5-32.
- Rose, L. A., Brenan, J.M. (2001). Wetting properties of Fe-Ni-Co-Cu-O-S melts against olivine: Implications for sulfide melt mobility. *Economic Geology*, 96(1), 145-157.
- Tornos, F., Velasco F., Hanchar J.M. (2017). The magmatic to magmatic-hydrothermal evolution of the El Laco deposit (Chile) and its implications for the genesis of magnetite-apatite deposits. *Economic Geology* 112, 1595–1628.
- Tornos, F., Velasco, F., Hanchar, J.M. (2016). Iron-rich melts, magmatic magnetite, and superheated hydrothermal systems: The El Laco deposit, Chile. *Geology*, 44(6), 427-430.
- Valley, P.M., Hanchar, J.M., Whitehouse, M.J. (2011). New insights on the evolution of the Lyon Mountain Granite and associated Kiruna-type magnetite-apatite deposits, Adirondack Mountains, New York State. *Geosphere*, 7(2), 357-389.
- Velasco, F., Tornos, F., Hanchar, J.M. (2016). Immiscible iron-and silica-rich melts and magnetite geochemistry at the El Laco volcano (northern Chile): Evidence for a magmatic origin for the magnetite deposits. *Ore Geology Reviews*, 79, 346-366.
- Williams, P.J., Barton, M.D., Fontbote, L., de Haller, A., Johnson, D.A., Mark, G., Marschik, R., Oliver, N.H.S. (2005) Iron oxide-copper-gold deposits: Geology, space-time distribution, and possible modes of origin: *Economic Geology 100th Anniversary Volume*, 371–406.
- Xavier, R.P., Wiedenbeck, M., Trumbull, R.B., Dreher, A.M., Monteiro, L.V., Rhede, D., Torresi, I. (2008). Tourmaline B-isotopes fingerprint marine evaporites as the source of high-salinity ore fluids in iron oxide copper-gold deposits, Carajas Mineral Province (Brazil). *Geology*, 36(9), 743-746.
- Zajacz, Z. (2015). The effect of melt composition on the partitioning of oxidized sulfur between silicate melts and magmatic volatiles. *Geochimica et Cosmochimica Acta*, 158, 223-244.

REVIEWERS' COMMENTS:

Reviewer #1 (Remarks to the Author):

The revised version of the manuscript by Hou et al. has addressed and clarified satisfactorily most of the comments in the previous version. There are only a few minor comments that can help polishing the manuscript:

Line 36. To my knowledge, there are no active IOA mines that recover phosphorous from apatite.

Line 35. The rocks are not always essentially free of silicate. In some mines there is abundant actinolite making sometimes up to 30% of the ore

Line 62. Difficult to understand. Do you mean common silicate minerals in the iron rich melt or in the original mafic silicate melt?

Line 77. Instead pyrite, use perhaps sulphur-rich minerals. As you quote, there is much more anhydrite than pyrite in many deposits.

Line 83. In your experiments, where the Ti comes from? From the (impure) magnetite used as the starting material?

Line 126. I would perhaps add "some" IOA deposits.

Line 144. Is SO₃ the dominant sulphur-bearing component? Is not SO₂?

Line 192. Fluorapatite instead apatite?

Line 205 and several other places. You systematically quote high redox conditions but one product of your experiments is fayalite, which should be stable under the QFM buffer. Any explanation? Perhaps I miss something.

1. Line 36. To my knowledge, there are no active IOA mines that recover phosphorous from apatite.

Yes we agree and removed 'P' from the sentence according to your suggestion and changed it into "These rocks, essentially free of silicates and sufficiently enriched in Fe to be recoverable, have been classified as Kiruna-type or iron oxide-apatite (IOA) deposits".

2. Line 35. The rocks are not always essentially free of silicate. In some mines there is abundant actinolite making sometimes up to 30% of the ore

This is true. If the rocks contain abundant actinolite, these ores could be explained by solidification of an immiscible ferrobaltic melt which separated from a Si-rich conjugate (Table 1). However, as pointed out in the introduction section, the formation of those ores free of silicate is more enigmatic, and this issue is the aim of this study. Thus, phase equilibria experiment on intermediate magmas have been conducted and the results show that increasing a_{H_2O} and fO_2 enlarges the two-liquid field thus allowing the Fe-Ca-P melt to separate easily from host silicic magma and produce silicate-free iron oxide-apatite ores.

3. *Line 62. Difficult to understand. Do you mean common silicate minerals in the iron rich melt or in the original mafic silicate melt?*

Above in the manuscript, we already mentioned “an iron-rich silicate melt”. Thus here “the melt” means “iron-rich silicate melt” This melt could be either formed by fractionation or by liquid immiscibility. Here we explain that IOA ores free of silicate “cannot simply result from differential crystal settling in an iron-rich silicate melt”.

4. *Line 77. Instead pyrite, use perhaps sulphur-rich minerals. As you quote, there is much more anhydrite than pyrite in many deposits.*

We changed in the revised manuscript.

5. *Line 83. In your experiments, where the Ti comes from? From the (impure) magnetite used as the starting material?*

Yes, the Ti comes from the (impure) magnetite used as the starting material

6. *Line 126. I would perhaps add “some” IOA deposits.*

We added “some” in the sentence.

7. *Line 144. Is SO₃ the dominant sulphur-bearing component? Is not SO₂?*

Yes, In IOA deposits, sulfur dissolves in the melt predominantly as S⁶⁺ as supported by the common presence of anhydrite.

8. *Line 192. Fluorapatite instead apatite?*

We changed apatite into fluorapatite.

9. Line 205 and several other places. You systematically quote high redox conditions but one product of your experiments is fayalite, which should be stable under the QFM buffer. Any explanation? Perhaps I miss something.

“High redox conditions” means QFM+3.3 in this study, in which no fayalite had been observed. All the fayalite crystals are produced in the experiments run under more reducing conditions, i.e., QFM+0.5.